# TSA Activates Pluripotency Factors in Porcine Recloned Embryos

**DOI:** 10.3390/genes13040649

**Published:** 2022-04-07

**Authors:** Tao Feng, Xiaolan Qi, Huiying Zou, Shuangyu Ma, Dawei Yu, Fei Gao, Zhengxing Lian, Sen Wu, Xuguang Du

**Affiliations:** 1State Key Laboratory of Agrobiotechnology, College of Biological Sciences, China Agricultural University, Beijing 100193, China; fengtao@caas.cn (T.F.); qixiaolan@caas.cn (X.Q.); gaofei2020019@cau.edu.cn (F.G.); swu@cau.edu.cn (S.W.); 2Institute of Urban Agriculture, Chinese Academy of Agricultural Sciences, Chengdu 610213, China; 3Embryo Biotechnology and Reproduction Laboratory, Institute of Animal Science, Chinese Academy of Agricultural Sciences, Beijing 100193, China; zouhuiying@caas.cn (H.Z.); ydw023@163.com (D.Y.); 4Department of Histo-Embryology, Genetics and Developmental Biology, School of Medicine, Shanghai Jiao Tong University, Shanghai 200336, China; sherrysyma@163.com; 5College of Animal Science and Technology, China Agricultural University, Beijing 100193, China; lianzhx@cau.edu.cn; 6Sanya Institute of China, Agricultural University, Sanya 572000, China

**Keywords:** reclone, pigs, *OCT4*/*SOX2* reporter

## Abstract

Animal cloning is of great importance to the production of transgenic and genome-edited livestock. Especially for multiple gene-editing operations, recloning is one of the most feasible methods for livestock. In addition, a multiple-round cloning method is practically necessary for animal molecular breeding. However, cloning efficiency remains extremely low, especially for serial cloning, which seriously impedes the development of livestock breeding based on genome editing technology. The incomplete reprogramming and failure in oocyte activation of some pluripotent factors were deemed to be the main reason for the low efficiency of animal recloning. Here, to overcome this issue, which occurred frequently in the process of animal recloning, we established a reporter system in which fluorescent proteins were driven by pig *OCT4* or *SOX2* promoter to monitor the reprogramming process in cloned and recloned pig embryos. We studied the effect of different histone deacetylase (HDAC) inhibitors on incomplete reprogramming. Our results showed that Trichostatin A (TSA) could activate pluripotent factors and significantly enhance the development competence of recloned pig embryos, while the other two inhibitors, valproic acid (VPA) and Scriptaid, had little effect on that. Furthermore, we found no difference in *OCT4* mRNA abundance between TSA-treated and untreated embryos. These findings suggest that TSA remarkably improves the reprogramming state of pig recloned embryos by restoring the expression of incompletely activated pluripotent genes *OCT4* and *SOX2*.

## 1. Introduction

Livestock breeding based on genome editing and animal cloning technologies could assign multiple desirable economic traits to one animal using less time. Compared to traditional breeding, this method shows a greater application prospect. Genome editing is highly efficient, thanks to the application of Clustered Regularly Interspaced Short Palindromic Repeats (CRISPR) and CRISPR-associated (Cas) technology [1,2,3,4]. Otherwise, due to the lack of high-quality porcine pluripotent stem cells that are capable of supporting multiple-rounds genome editing, methods of multiple-round cloning are frequently used in animal breeding. But the efficiency of animal cloning remains low, especially for serial recloning. The activation failure of key pluripotent genes is a major problem in reprogramming mediated by Yamanaka factors (induced pluripotent stem cells technology) [5,6] or by mammal oocytes (animal cloning) [7,8]. To improve reprogramming efficiency, the effect of histone deacetylase (HDAC) inhibitors on reprogramming efficiency has been studied intensively in induced pluripotent stem cells (iPSCs). Previous results showed that HDAC inhibitors, such as Trichostatin A (TSA) and valproic acid (VPA), could be used to benefit the generation of iPSCs [9,10]. However, similar studies were difficult to carry out in cloned embryos, because quite a few cells were present in early developmental embryos, especially in cloned embryos, which made it very difficult to detect translation changes. Further, Macfarlan et al. suggested that key pluripotent genes were regulated post-transcriptionally [11], and the data based on mRNA quantitation might not be able to reflect the actual level of these proteins faithfully. Therefore, a reliable and real-time system to monitor the dynamics of gene expression in protein-level synthesis will be a powerful tool to resolve the problem.

EGFP linked with the *OCT4* promoter (OCT4-EGFP) has been used as a reporter to monitor the expression dynamics of *OCT4* during early embryonic development and the somatic reprogramming process in mice [12,13]. The expression profile of EGFP mimics the expression of endogenous *OCT4* in human embryonic stem cells (ESCs) carrying the transgene OCT4-EGFP [14], and this reporter system has also been used in other mammals to study various aspects of gene expression [15,16,17,18,19]. Therefore, a reporter system with fluorescent protein driven by the promoter of a specific pluripotency gene, such as *OCT4* or *SOX2*, is a convenient tool to study the real-time translational dynamics of these genes and help reveal the reprogramming mechanism underlying early embryonic development and cloning.

Here, we aimed to determine whether inhibitors of HDACs could activate pluripotent factors during the reprogramming mediated by pig oocytes. Our results based on the OCT4-EGFP/SOX2-tdTomato reporter systems suggest that, after the TSA treatment, pluripotent gene expression and development competence could be enhanced, especially for recloned embryos. Furthermore, no variation could be detected in mRNA abundance, which suggests that the reprogramming enhancement effect by TSA could be carried out through the post-transcriptional regulation mechanism.

## 2. Materials and Methods

All chemicals were purchased from Sigma Co. (St. Louis, MO, USA), unless otherwise stated. Animal experiments in this study were approved by the Animal Welfare Committee of China Agricultural University (SKLAB-2012-11).

### 2.1. Construction of Reporter Vectors

The genomic DNA extracted from mini pigs with a DNA extraction kit (Takara Bio, no. 9762, Tokyo, Japan) was used as template for PCR amplification of *OCT4* and *SOX2* promoters by Herculase DNA polymerase (Agilent, no. 600675, Santa Clara, CA, USA). The cycling conditions were: 2 min at 94 °C, 28 cycles of 94 °C for 20 s 58 °C for 20 s, 72 °C for 4 min, and 72 °C for 5 min. The primers used are listed in Appendix A. The DNA fragment of promoters was subcloned to the pZGs/pZTs vector [20] to obtain pPB-pOCT4-EGFP and pPB-pSOX2-tdTomato vectors. The fragments of pOCT4-EGFP and pSOX2-tdTomato were isolated and subcloned into a non-transposon vector pNP-EF1aEGFP-SV40puro to produce pNP-OCT4-EGFP (OCT4-EGFP) and SOX2-tdTomato (SOX2-tdTomato) by *Spe* I/*BsrG* I. Both of the vectors were linearized by *Asc* I before nucleofection.

### 2.2. Generation of Porcine Embryonic Fibroblasts (PEFs) Carrying the OCT4-EGFP or SOX2-tdTomato Reporter

The porcine embryonic fibroblasts (PEFs) used here were isolated and preserved by our laboratory. The PFFs were isolated from day-28 porcine embryos of Bama mini pigs. After passaging 1–2 times, 4 × 10^4^ PEFs were seeded into one well of a 6-well dish. The procedure of transfection was performed on Amaxa Nucleofector (Lonza, no. VCA1003, Basel, Switzerland) using program A-024 and transfected cells were placed in 100-mm dishes. After two days of culture, selective media with 1 μg/mL puromycin (Invitrogen, no. A1113803, Carlsbad, CA, USA) were used to get drug-resistant cell colonies for an additional 8–10 days. Drug-resistant cell colonies were picked with the cloning ring and propagated in a 48-well culture plate. When reaching 90% confluence, cells were trypsinized (0.05% trypsin) and passaged into a 24-well plate, and then to 12-well dishes to be cryopreserved and screened for transgene by PCR immediately. For PCR screening, 30% cells of each clone were put in 20 μL of HotShot lysis buffer, boiled at 90 °C for 15 min, and neutralized with 20 μL of 40 mM Tris-Cl [21]. After vortexed thoroughly, the mixture was used as the template for PCR. The PCR conditions were specified as follows: 4 min at 95 °C; 30 cycles of 95 °C for 30 s, 58 °C for 50 s, and 72 °C for 30 s; and 72 °C for 5 min.

### 2.3. Somatic Cell Nuclear Transfer

Porcine ovaries were obtained from a local slaughterhouse and transported to the laboratory in 0.9% NaCl at 35–38 °C within 2.5 h. Follicular fluid was aspirated from medium-sized (3–6 mm) follicles by using a 12-gauge needle attached to a vacuum pumping system. Cumulus oocyte complexes (COCs) with several layers of cumulus cells were selected and washed two times in the in vitro maturation (IVM) medium, which contained 780 mL TCM-199 (M-2154), 20,000 IU PMSG (Ningbo Second Hormone Factory, no. 160310, Ningbo, China), 10,000 IU hCG (Ningbo Second Hormone Factory, no. 160119, China), 40 mg Glutamine (G8540), 146 mg Gentamicin (G1264), 10% (*v*/*v*) follicular fluid, and 10% (*v*/*v*) cow serum with an adjusted osmotic pressure of 280 ± 8 mOsm, and then 50–70 COCs were cultured in 500 μL IVM covered with mineral oil (M5904), which was pre-equilibrated at 38.5 °C, 5% CO_2_ for more than 4 h. After 42–43 h of culture, cumulus cells were removed by pipetting COCs for about 1 min in 0.1% hyaluronidase (H3506). Oocytes with an intact plasma membrane were selected and enucleation was performed by using a pipette with a 20–25 μm internal diameter under an inverted microscope (Nikon, SMZ800, Tokyo, Japan) equipped with a micromanipulator (Narishige, NT88-V3, Tokyo, Japan). A single donor cell was injected into the perivitelline space of an enucleated oocyte. The fusion of donor cell with recipient oocyte and activation of reconstructed embryos was carried out by two direct current pulses of 150 V/mm for 100 μs in F/A buffer (0.25 M mannitol, 0.1 mM CaCl_2_ 2H_2_O, 0.1 mM MgCl_2_-6H_2_O, 0.5 mM Hepes, 0.01 g/100 mL polyvinyl alcohol) using an electrical transfection instrument (BLS, CF-150B, Budapest, Hungary). Fused embryos were cultured in 5 μg/mL cytochalasin B and 10 μg/mL cyclohexane-supplemented PZM-3 for 4 h. Then, 50–80 embryos were transferred to 500 μL PZM-3 in 4-well dish and covered with mineral oil. The embryos were cultured at 38.5 °C in 5% O_2_, 5% CO_2_, and 90% N_2_ with maximum humidity. Blastocyst formation was evaluated six days after SCNT.

### 2.4. Embryo Transfer

About 600 one-cell embryos were surgically transferred into two surrogate gilts one day after the start of estrus, and the pregnancy status of the surrogates was diagnosed by ultrasonography one month later.

### 2.5. Establishment of PEFs

Thirty days after embryo transfer, pregnant surrogate mothers were euthanized and fetuses were collected by using an aseptic procedure. After removal of the head, viscera, and extremities, the fetuses were thoroughly rinsed three times with phosphate-buffered saline (PBS). Then the tissues were minced into pieces using an ophthalmic scissor and incubated in 10 mL 0.1% trypsin-EDTA solution with shaking at 38 °C for 5 min. The cell pellets were suspended with 20 mL DMEM (Gibco, no. 11960, Carlsbad, CA, USA) plus 15% FBS (Gibco, 16141, Carlsbad, CA, USA), centrifuged at 1000 rpm for 5 min, and resuspended in DMEM supplemented with 15% FBS, 0.5 mM GlutaMAX (Gibco, no. 35030, Carlsbad, CA, USA), penicillin/streptomycin (Gibco, no. 15140, Carlsbad, CA, USA), and nonessential amino acid (Gibco, no. 11140, Carlsbad, CA, USA). When PEFs reached 90% confluence, they were trypsinized into single cells and then cryopreserved in cryovials, with about two vials for each fetus. After cooling to −80 °C at 1 °C/min using Nalgene Mr. Frosty^TM^ freezing container (Thermo Fisher, Waltham, MA, USA), the vials were then stored in liquid nitrogen.

### 2.6. HDAC Inhibitors Treatment

Immediately following activation, embryos were treated with one of the histone deacetylase (HDAC) inhibitors: 40 nM Trichostatin A (TSA), 1 mM valproic acid (VPA) or 500 nM Scriptaid for 4 h, in PZM-3 supplemented with 5 μg/mL cytochalasin B and 10 μg/mL cyclohexane, and were incubated in PZM-3 supplemented with 5 μg/mL cytochalasin B and 10 μg/mL cyclohexane for 4 h. Then, embryos were transferred into PZM-3 containing TSA, VPA, or Scriptaid at the same concentrations as previously described for 20 h. Activated embryos were cultured for 4 h in PZM-3 supplemented with 5 μg/mL cytochalasin B, 10 μg/mL cyclohexane and the same volume of dimethylsulfoxide (DMSO, D2650) as used in treatment groups to serve as a control. After these treatments, the embryos were washed twice in PZM medium and cultured in PZM-3 for further development.

### 2.7. Immunostaining

Embryos were rinsed in PBS supplemented with 1% (W/V) BSA, fixed in freshly prepared 4% paraformaldehyde for 5 min at room temperature (RT), and permeabilized in PBS containing 0.2% Triton X-100 for 15 min at RT. Embryos were then blocked with PBS containing 1% BSA (blocking buffer) for 1 h and incubated with primary antibodies diluted 1:100 in blocking buffer overnight at 4 °C (OCT4 antibody, SC 5729, Santa Cruz Biotechnology, Dallas, TX, USA). After overnight incubation, embryos were washed with blocking buffer three times and then incubated with secondary IgG antibodies conjugated with Cy5 (Jackson Immuno Research, AAT-16855, West Grove, PA, USA). Nuclei were counterstained with 4’,6-diamidino-2-phenylindole (DAPI) before mounting onto the slides. The slides were observed under a fluorescence microscope (Leica, DM IL LED, Wetzlar, German).

### 2.8. Quantitative Real-Time PCR (q-PCR)

For each group, total RNA was extracted from 50 Day 4 (E4) cloned or recloned embryos using Cell-to-cDNA^TM^ II Cell Lysis Buffer (Thermo Fisher, AM8723, Waltham, MA, USA). cDNA was generated directly using M-MLV reverse transcriptase (Promega, M530B, USA) following the manufacturer’s instruction. qPCR was carried out on LightCycler^®®®^ 480 (Roche, Basel, Switzerland) using LightCycler 480 SYBR Green I Master (Roche, no. 4887352001, USA) with the program: 94 °C for 2 min, 30 cycles of 94 °C for 30 s; 60 °C for 30 s; 72 °C for 30 s; and 72 °C for 5 min. The primers used are listed in Appendix A. Gene expression was normalized to the corresponding housekeeping gene *GAPDH* [22,23].

### 2.9. Bisulfite Sequencing

Bisulfite treatment of genomic DNA was carried out by using MethylDetector^TM^ Bisulfite Modification Kit (Active Motif, no. 55001, Carlsbad, CA, USA) according to the manufacturer’s instruction. Briefly, the genomic DNA was treated with sodium bisulfite to perform conversion of unmethylated cytosines to uracils with conversion buffer. Then, nested PCR (primers seen in Appendix A) was carried out to amplify the converted DNA by using Hot Start Taq^TM^ Polymerase (Takara Bio, R028A, Tokyo, Japan) with a program of 94 °C (5 min) and 45 cycles of 94 °C (30 s), 55 °C (30 s), 72 °C (1 min), and 72 °C (10 min). The PCR products were purified with an Agarose Gel DNA Purification Kit (Takara Bio, DV805A, Tokyo, Japan). The purified fragments were then cloned into pMD18-T Vector (Takara Bio, no. 6011, Tokyo, Japan) for sequence analysis.

### 2.10. Statistical Analysis

All experiments were repeated at least three times. Data for blastocyst rate and EGFP-blastocyst rate were presented as mean ± standard deviation (SD) and analyzed by chi-square test with SPSS(Version 19.0, IBM, Armonk, NY, USA). *p* values smaller than 5% were considered to be statistically significant.

## 3. Results

### 3.1. Monitoring the Reprogramming Process in Pig SCNT Embryos by a Novel OCT4-GFP/SOX2-tdTomato Reporter Systems

To monitor the status of pluripotent genes during the early development stage, we first established two reporter systems (OCT4-EGFP, SOX2-tdTomato). *OCT4* promoter fragments were amplified from the pig genomic DNA (primers listed in Appendix A), and then cloned into pNP-OCT4-EGFP/SOX2-tdTomato (OCT4-EGFP, pOG; SOX2-tdTomato, pST) vector (Figure 1A). PEFs were electro-transfected with pOG and pST vectors, selected with puromycin, and cultured for 10 days. Then, SCNT was carried out to derive embryos from transgene positive PEFs. EGFP/tdTomato fluorescence was monitored from one-cell stage to blastocysts. We found that both OCT4-EGFP and SOX2-tdTomato were successfully activated in the cloned embryos but not in non-transgenic control embryos or transgenic PEFs (Figure 1B).

To confirm the expression of endogenous *OCT4* in transgenic reporter-embryos when OCT4-GFP or SOX2-tdTomato was activated, immunostaining for endogenous OCT4 was conducted. The results showed that endogenous *OCT4* was turned on in reporter lines after SCNT as well as in parthenogenetic embryos (Figure 1C). The results indicated that the reporter system works well in monitoring pluripotent gene expression after SCNT.

### 3.2. Identification of TSA as an Enhancer of Reprogramming Efficiency

We transferred cloned embryos containing pOG vector into surrogate mothers, and 15 PEF cell lines were established from 30-days fetuses. The transgenic cell lines were used as donors for recloning.

In the second-round cloning, we observed a remarkable reduction in the number of EGFP fluorescent blastocysts (Figure 2A,B), delayed activation of OCT4-EGFP in most embryos, and a higher level of methylation of CpG islands of *OCT4* promoter compared to that of the first-round cloning (Figure 2C). To determine the effect of HDAC inhibitors on the expression of the *OCT4* gene in recloned embryos, we treated the recloned embryos with TSA, VPA, and Scriptaid. The results show that TSA treatment significantly activated the expression of *OCT4*, while VPA and Scriptaid had only a slight improvement on that (Figure 2D). Furthermore, we tested the effect of TSA using cloned embryos labeled with the pST reporter. The results suggest that TSA performed a similar effect on pluripotent gene *SOX2* (Figure 3A). These results suggest that TSA could rescue, at least partly, the incomplete activation of pluripotent genes in pig cloned and recloned embryos.

### 3.3. Development Enhancement Mediated by TSA in Recloned Embryos

To investigate the effect of TSA on the developmental competence of embryos in the process of somatic nuclear reprogramming mediated by oocytes in pigs, cloned pig embryos derived from transgene positive cell clone (#1) were treated with or without TSA. Results show that the percentage of EGFP positive blastocysts in the TSA treatment group was significantly higher than that of the control group (Table 1). Furthermore, results from embryos with SOX2-tdTomato reporter showed that TSA treatment also significantly activated *SOX2* in cloned embryos (Figure 3A).

To investigate the effect of TSA on the development of recloned embryos, recloned embryos were treated with or without TSA after fusion and activation. Similar to the results in the first round of SCNT experiments, TSA was able to significantly promote the development ability of recloned embryos, and the percentage of EGFP-blastocysts in the TSA treatment group was about three times that of the control group (Table 2). In control groups, the EGFP-blastocyst rate in second-round SCNT was lower than that of first-round SCNT (Figure 2B, *p* < 0.05). However, data in Table 1 and Table 2 showed that, after TSA treatment, EGFP-blastocyst rates between first- and second-round cloning experiments were similar (57.0 ± 9.4% vs. 51.1 ± 4.3%). Specifically, the expression of OCT4-EGFP in most TSA-untreated embryos was delayed until the morula stage. After TSA treatment, there was no delay in the expression of OCT4-EGFP, and the EGFP expression was observed in embryos as early as 4/8-cell stage (E3) (Figure 3B). These results indicated that TSA treatment could activate pluripotent genes and enhance the developmental competence of cloned and recloned pig embryos.

At last, to find out whether TSA could increase transcription of the two pluripotent genes, q-PCR was performed in recloned embryos (E4) treated with or without TSA to determine *OCT4* and *SOX2* expression levels. No differences were observed in either *OCT4* or *SOX2* expression in recloned embryos between the respective treatment group and its untreated control (Figure 4). The results indicate that TSA treatment does not affect the expression of endogenous genes *OCT4* and *SOX2* at the mRNA level.

## 4. Discussion

Improvement of animal cloning efficiency is essential for the practical application of animal breeding based on transgenic and genome editing technologies. In this study, we investigated the effects of HDAC inhibitors on the reprogramming of PEFs after SCNT. The results showed that (1) our reporter system could be used to faithfully monitor the dynamic of pluripotent gene *OCT4* and *SOX2* during SCNT-mediated reprogramming events, and (2) HDAC inhibitors TSA, other than Scriptaid and VPA, could rescue reduced reprogramming efficiencies of recloned porcine embryos.

Although injection of Cas9 protein and single-guide RNA (sgRNA) into fertilized eggs has been successfully applied to produce gene-targeted animals, this method frequently resulted in mosaicism of the targeted gene modification [24,25]. Further breeding is necessary for the achievement of homozygotes with identical genotype and phenotype. The SCNT method can address this issue. In practice, multiple rounds of cloning are important to animal cloning, due to the lack of high-quality pluripotent stem cells in large animals. But, currently, the efficiencies of serial cloning in large animals are extremely low. Therefore, we established a reporter system to investigate the status of pluripotent genes during SCNT. Both immunostaining and fluorescent reporting results showed that OCT4 expression could be detected in both trophoblasts and inner cell mass (Figure 1C), which is in contrast with the ICM-specific expression pattern of OCT4 in mice, but similar to that in cattle [26]. In general, these results demonstrated that our reporter system could be used to faithfully monitor the dynamic of pluripotent gene *OCT4* and *SOX2* during SCNT-mediated reprogramming events.

The expression levels of pluripotent genes, especially *OCT4* and *SOX2*, highly correlated with the efficiency of reprogramming. Our early studies suggested that a close correlation exists between the partially reprogrammed state of porcine iPSCs and the inactivation of endogenous *OCT4* [27]. Previous reports showed that epigenetic errors are the main reasons for developmental failures [28,29], and these errors could accumulate during serial cloning [30]. Therefore, the recloning method provides us with a better model to study incomplete reprogramming. Here, we performed cloning and recloning using pig oocytes; the results showed that pluripotent factors *OCT4* and *SOX2* of most reconstructed embryos were activated at the 4-cell stage during first-round cloning, but in second-round cloning, their activation in most embryos was largely delayed until the morula stage. These delays may be explained by the accumulation of epigenetic or genetic errors that occurred during serial cloning. And these delays could be the main hindrance for blastocyst formation or efficient recloning. Our data demonstrated that TSA could help to relieve this hindrance, while VPA and Scriptaid had little effect on that.

Besides improving the expression delay of *OCT4*, we tested the effect of TSA treatment on the expression of pluripotent genes *OCT4* and *SOX2*. Our results of q-PCR show that TSA treatment has no significant effect on the expression of *OCT4* and *SOX2* (Figure 4). Studies of TSA mainly focused on its role in acetylation regulation. It has been proved that TSA treatment could decrease the acetylation level of SCNT embryos [31,32,33]. As an epigenetic modification, low-level acetylation facilitates gene expression and is considered to be the main determinant for successful reprogramming. We failed to detect a significant difference between TSA-treated embryos and controls, and this is consistent with previous results in cloned mouse and pig blastocysts [31,34]. Given that the expression of pluripotency genes *OCT4* and *SOX2* is regulated strictly and dynamically in early development [35], those data do not support the conclusion that TSA does not affect the expression of *OCT4* and *SOX2*. This is because results in this and other studies were obtained at a single time point during the preliminary development of embryos (E4 or blastocyst stage) [31,34]. To comprehensively investigate the effect of those HDAC inhibitors on the pluripotent gene expression, samples at multiple time points should be collected to perform tests.

In this study, we established a reporter system to monitor the status of pluripotent genes during SCNT and found that TSA promotes the development potential of cloned pig embryos, especially for recloned embryos. These findings help us justify the potential of using small molecules like TSA to improve somatic cell reprogramming, especially in large animals. All these results are beneficial for the development of livestock molecular breeding.

## Figures and Tables

**Figure 1 genes-13-00649-f001:**
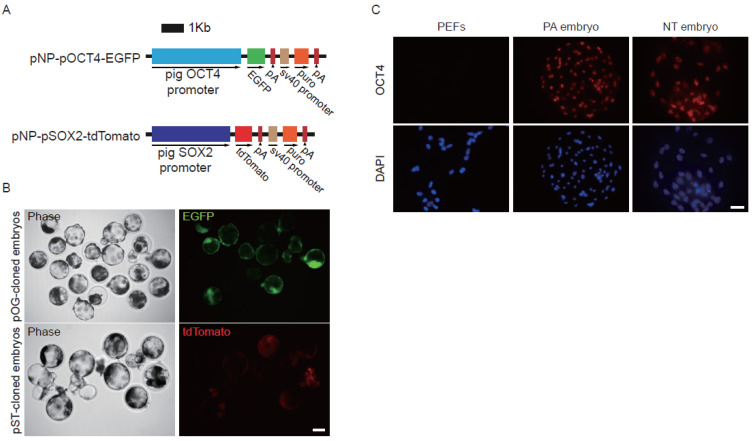
Expression of the OCT4-EGFP and SOX2-tdTomato reporters in the cloned embryos. (**A**) Schematic representation of the OCT4-EGFP and SOX2-tdTomato vectors construction. (**B**) The reporter vectors expressed in SCNT embryos. (**C**) Detection of OCT4 protein in the EGFP positive cloned embryos harboring the pOG vector. PEFs and parthenogenetic (PA) embryos were used as the negative and positive controls. The scale bar represents 100 μm, 50 μm in (**B**,**C**), respectively.

**Figure 2 genes-13-00649-f002:**
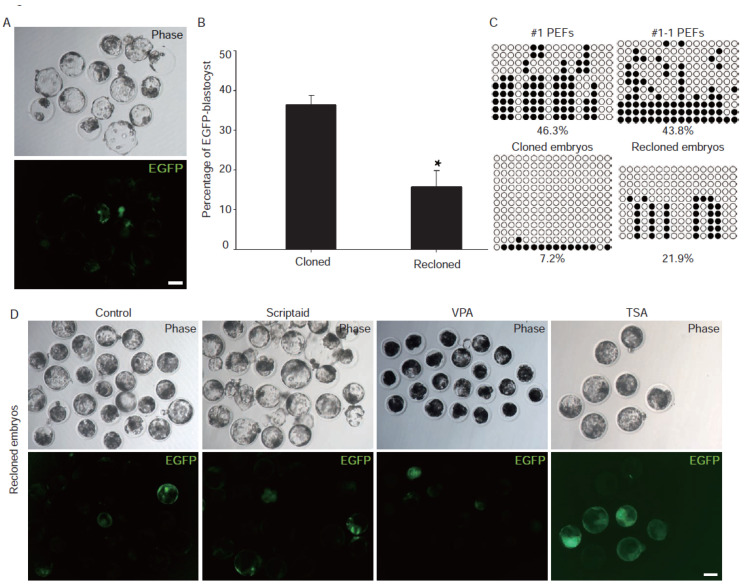
Identification of TSA as an enhancer of developmental efficiency. (**A**) Reporter pOG was downregulated in recloned embryos. The scale bars represent 100 μm. (**B**) Comparison of the EGFP-blastocyst rate between cloned and recloned embryos. Asterisks indicate significant differences (*t*-test, *p* < 0.05). (**C**) Methylation status of the *OCT4* promoter in PEFs (upper panel), cloned, and recloned embryos (lower panel). (**D**) Effect of TSA, VPA, and Scriptaid on recloned embryos. The upper and lower panels represent embryos in the bright and fluorescent fields, respectively. Recloned embryos treated with equivalent DMSO were used as a negative control. The scale bar represents 100 μm.

**Figure 3 genes-13-00649-f003:**
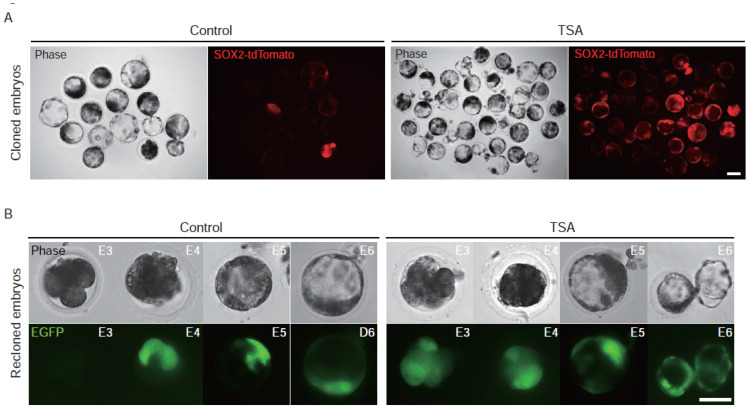
TSA-activated pluripotent factors in pig SCNT-embryos. (**A**) Activation of *SOX2* mediated by TSA in cloned embryos. Cloned embryos were reconstructed by using PEFs harboring the transgene pSOX2-tdTomato. Embryos treated with equivalent DMSO were used as controls. The scale bar represents 100 μm. (**B**) Expression pattern of reporter pOG in recloned embryos with or without TSA treatment. Embryos were observed at 4-/8-cell stage (E3), morula stage (E4), early blastocyst stage (E5), and late blastocyst stage (E6). The cloned embryos treated with equivalent DMSO were used as controls. The scale bar represents 50 μm.

**Figure 4 genes-13-00649-f004:**
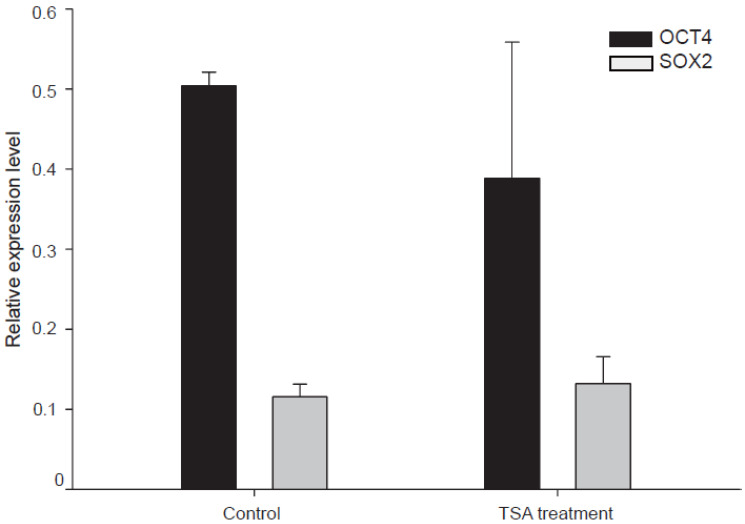
Analysis of the mRNA abundance of *OCT4* gene. Relative transcription of *OCT4* and *SOX2* in embryos with TSA treatment. The analysis was performed on day four of embryo development. Control embryos were recloned embryos treated with equivalent DMSO. The transcription values of *OCT4* and *SOX2* were normalized with corresponding *GAPDH* transcription values.

**Table 1 genes-13-00649-t001:** Effect of TSA on the reprogramming efficiency of cloned embryos.

Treatment	No. of SCNT-Embryos	No. of Blastocysts (%)	No. of EGFP-Blastocysts (%)
Control	337	47 (13.9 ± 1.3) ^a^	16 (36.4 ± 2.4) ^a^
TSA	240	50 (20.9 ± 1.5) ^b^	28 (57.0 ± 9.4) ^b^

Blastocyst rate: number of blastocysts/number of reconstructed embryos. EGFP-blastocyst rate: number of EGFP-blastocysts/number of blastocysts. Within the same column, different superscripts show significant differences, *p* < 0.05.

**Table 2 genes-13-00649-t002:** Effect of TSA on the reprogramming efficiency in recloned embryos.

Treatment	No. of SCNT-Embryos	No. of Blastocysts (%)	No. of EGFP-Blastocysts (%)
Control	350	46 (13.2 ± 0.9) ^a^	8 (15.8 ± 4.1) ^a^
TSA	366	93 (25.5 ± 4.2) ^b^	47 (51.1 ± 4.3) ^b^

Blastocyst rate: number of blastocysts/number of reconstructed embryos. EGFP-blastocyst rate: number of EGFP-blastocysts/number of blastocysts. Within the same column, different superscripts show significant differences, *p* < 0.05.

## Data Availability

Not applicable.

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
