# Peer review of "TSA Activates Pluripotency Factors in Porcine Recloned Embryos"

_genes, 2022, doi:10.3390/genes13040649_

Round 1

Reviewer 1 Report

The work shows interesting results. I found some language mistakes in the article, please double-check the manuscript in this regard. Due to the lack of a line counter, the proposed corrections will be based on sections and paragraphs.

Material and methods: Where were the fibroblasts derived from? Were these fibroblast commercial lines or were they derived for this experiment?

Somatic cell nuclear transfer subsection: there is the wrong font used for "performed"

Establishment of PEFs subsection: the degree symbols are underlined

Quantitative real-time PCR subsection: Is GADPH a properly selected housekeeping gene? Are there any literature data or have tests been carried out?

Bisulfite sequencing subsection: Did you mean "MethylDetector" and "Hot Start"?

Results: Many of the sentences fit more into the discussion section. I propose that sentences covering: the last four and a half lines of the first subsection (Monitoring reprogramming…); the first three and a half lines of the second subsection (Identification of TSA…); a fragment of a sentence in the fourth line of the third subsection (Development enchantment ...) referring to other studies; the last sentence of the same paragraph in this subsection; and the last two sentences of the fourth subsection (RNA expression ...) transfer to the discussion section and create appropriate paragraphs on their basis.

Discussion: The section, in my opinion, does not refer too much to the results obtained in this work and their comparison with the available literature. I suggest you expand it, or combine it with the results section and create a new conclusion section.

Table S1: The table is missing accession numbers for the reference sequences from which the primer sequences were designed. Please provide them. Additionally, the fonts used are not uniform, please improve.

Reviewer 2 Report

The authors used OCT4 or SOX2 reporter system to monitor the reprogramming process in pig cloned and re-cloned embryos. They also used three histone deacethyltransferase inhibitors (Trichostatin A, valproic acid and Scriptaid) and found that Trichostatin A could activate pluripotent factors and significantly enhance the development competence of pig re-cloned embryos.  Overall, the results were enough to support the conclusions, especially even they found OCT4/SOX2 reporter genes were up-regulated in TSA-treated groups, the endogenous OCT4/SOX2 were not different. This is an important point to discuss how to improve cloning efficiency in SCNT embryos. However, I’d like to indicate some minor points below.

Abstract
-To over come this issue which occurred frequently in the process of animal recloning, we established…

Introduction
-p.2 L.2 [8-10] -> [8, 9]
-p.2 L.15 [11] -> [10]
-p.2 L.11 [12, 13] -> [11, 12]
-p.2 L.13 [14] -> [13]
-p.2 L.14 [15-18] -> [14-18] Are these correct?

Materials and 
-Please add the permission number of animal experiments.
-p.3 L.18 “performed” -> change the font size

-Results
p.5 L.39-40 (the last sentence of the second section) This seems to be over-interpretation, as only OCT4/SOX2 expression changes were observed at this time. No blastocyst rate, successful rate of cloning were observed (next section). Please re-write this sentence.

-References
7. Metoba -> Matoba
(Please check the reference format again)

Fig.1A pNS-pOCT4-EGFP -> pNP-pOCT4-EGFP?
pNS-pSOX2-tdTomato -> pNP-pSOX2-tdTomato?
Fig.2B pS2-cloned embryos - pST-cloned embryos?

Table S1. “For methylation-specific PCR” -> “For bisulfite PCR” (not methylation-specific PCR)
What do “pOUT4-S” and “pOUT4-A” mean? Why are they not capital? 
